# Dense Correspondences between Human Bodies via Learning Transformation Synchronization on Graphs

**Xiangru Huang**
The University of Texas at Austin
2317 Speedway, Austin, Texas
xiangruhuang816@gmail.com

**Haitao Yang**
The University of Texas at Austin
2317 Speedway, Austin, Texas
yanghtr@outlook.com

**Etienne Vouga**
The University of Texas at Austin
2317 Speedway, Austin, Texas
evouga@cs.utexas.edu

**Qixing Huang**
The University of Texas at Austin
2317 Speedway, Austin, Texas
huangqx@cs.utexas.edu

## Abstract

We introduce an approach for establishing dense correspondences between partial scans of human models and a complete template model. Our approach's key novelty lies in formulating dense correspondence computation as initializing and synchronizing local transformations between the scan and the template model. We introduce an optimization formulation for synchronizing transformations among a graph of the input scan, which automatically enforces smoothness of correspondences and recovers the underlying articulated deformations. We then show how to convert the iterative optimization procedure among a graph of the input scan into an end-to-end trainable network. The network design utilizes additional trainable parameters to break the barrier of the original optimization formulation's exact and robust recovery conditions. Experimental results on benchmark datasets demonstrate that our approach considerably outperforms baseline approaches. Our result, pretrained model and code are publicly available at https://github.com/xiangruhuang/HumanCorresViaLearn2Sync.

## 1 Introduction

We introduce a novel approach for computing dense correspondences between partial scans of human subjects and a complete template model. The correspondences computed with our approach are more accurate than the state of the art, and are robust to input noise, pose of the scanned subject, and variations in body shape and other non-rigid deformations of the scan relative to the template. Our method is also efficient, capable of matching a 3 000-point partial scan to a 6 890-vertex template in under 66 ms—suitable for real-time tracking applications running at 15 fps.

Algorithms for computing dense correspondences typically combine two key ideas. *Local geometric features* are used to match points on the source to candidate points on the target. Only a sparse set of prominent feature points can be reliably matched using these shape cues; to compute high-quality dense correspondences insensitive to noise, pose, and global symmetries (such as human bilateral symmetry), priors are enforced in the form of *global regularization constraints* (such as preservation of pairwise distances or orientation [23, 46]).

While early works used hand-crafted shape descriptors to identify geometric features [23], recent approaches [51, 15, 42, 31, 13, 17] have demonstrated that deep neural networks excel at this task. However, the use of neural networks introduces a new challenge: there are many possible choices for how to represent the set of dense correspondences [3, 2, 19, 29, 38, 33], and this choice of data representation is critical, as it dictates which neural network architectures can be used and what types of regularization constraints can be enforced.

Our approach is motivated by the observation that the human body behaves, to good approximation, as an articulated rigid body; poses can be explained by deformations that are almost rigid locally [44], away from the joints. Similarly, mappings between different subjects are also nearly piecewise-rigid, where variation in body shape can be explained by a small and almost-constant deviation away from a rigid transformation at the local scale. Our key idea, then, is to use *local rigid transformations* as the data representation for correspondences: we associate to each source point a rigid transformation that characterizes the local source-to-target deformation near that point, and formulate dense correspondence computation as synchronization of these local transformations over the graph of nearest neighbors on the input scan.

We initialize these local rigid transformations by fitting dense correspondences derived from a learned shape descriptor, and then perform transformation synchronization to jointly optimize the local rigid transformations. Motivated by recent trends in formulating iterative optimization procedures as recurrent neural networks [58, 40, 24, 22, 50], and recent advances in graph convolution [57, 52], we formulate transformation synchronization as a recurrent neural network. Driven by a robust optimization formulation of transformation synchronization, an essential advantage of our approach is that it automatically enforces smoothness of correspondences, and these correspondences are locally explainable in terms of articulated rigid motion of the source away from the target. The entire network (including both transformation initialization and synchronization) admits end-to-end training.

We evaluate the proposed technique on multiple benchmark datasets. Our correspondences have mean accuracy of 1.90cm, and 4.81cm on FAUST correspondence task [6], and SHREC19 [34], which are 26.7% and 17.6% better than using the state-of-the-art SMPL model [33] for template matching. When applying our approach to match complete shapes, i.e., by integrating correspondences between simulated scans and the template (c.f. [51]), our approach also considerably outperforms existing state-of-the-art [11, 51, 13, 10].

## 2   Related Work

Computing dense correspondences between geometric objects has an extensive history, and we refer the reader to a few standard surveys [23, 46] on this topic. Since the focus of this paper is on developing priors on correspondences and on the underlying shape deformation, in order to exploit them for global regularization, we classify relevant works based on different regularization strategies.

**MRF inference.** A standard paradigm for dense correspondence computation is to solve an MRF inference problem [3, 55, 8], where the unitary potentials model dense correspondence scores, and high-order potentials score consistency between multiple correspondences. A limitation of these approaches is their computational cost. To improve performance many methods take advantage of relaxed formulations including spectral relaxations [27, 19] and convex relaxations [55, 8]. In contrast to MRF inference, our approach's key advantage is the use of local transformations as latent variables to model consistency between nearby correspondences. Our approach is flexible in modeling various constraints, such as the smoothness and articulated rigid structure of the correspondences.

**The functional map framework.** Functional maps [38] and their subsequent refinements [41, 35] provide a flexible and efficient mapping framework between linear function spaces on two surfaces. In this framework, mappings between surfaces are represented as linear transformations. Such a linear algebraic structure enables many applications, such as solving non-rigid puzzles [32] and joint analysis of object collections [48, 49, 21, 20]. Functional maps can be easily connected with descriptor computation towers [31, 15, 42, 11], enabling end-to-end training in both supervised [31] and unsupervised [15, 42] manner. Challenges of the functional map framework include difficulty in recovering pointwise correspondences and with enforcing *extrinsic* constraints such as articulated deformation of the source away from the target. Our approach overcomes both of these challenges.

**Parameterization-based techniques.** Another category of methods leverages intermediate parameterization domains [29, 25, 1, 26]. Key advantages of this approach include ensuring injectivity of the mapping and generating very dense (i.e., continuous) correspondences. However, these approaches usually assume that the source and target are manifold meshes, and provide no help in enforcing priors on the extrinsic deformation (such as piecewise-rigid structure.)

**Template-based matching.** When matching a partial scan to a complete 3D model, a popular approach is to perform template matching. Early works [28, 14, 60, 37, 53, 54, 59] explicitly model shape deformations and leverage extrinsic distance metrics for matching. Thanks to advances in generative modeling, recent methods [30, 13, 17] learn models of shapes and formulate template matching as optimizing parameters in the learned latent space. In contrast to black-box modeling of

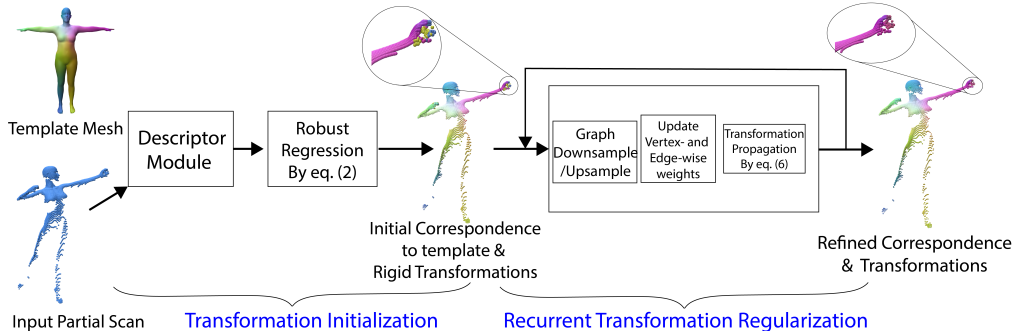

Figure 1: Our network combines a transformation initialization module and a transformation regularization module. The initialization module fits rigid local transformations to dense correspondences derived from a descriptor module. The regularization module is derived from a robust transformation synchronization formulation to rectify initial transformations. The entire network is end-to-end trainable.

shape deformations, our recurrent module is based on a principled optimization-based formulation of deformation regularization, and our method explicitly models smooth and articulated deformation.

**Recurrent networks for dense correspondences.** For optimization problems that admit iterative solvers [58], a recent trend in machine learning is to convert them into recurrent networks. This strategy has been adapted for object matching recently. One recent paper [40] presents a recurrent procedure to compute weighted correspondences for estimating the fundamental matrix between two images. Another [24] uses recurrent networks to progressively compute dense correspondences between two images. The network design is motivated by the traditional variational procedure for non-rigid image registration. Huang et al. [22] introduced a recurrent network for synchronizing transformations among multiple point clouds. Finally, PRNet [50] presented a recurrent module for predicting keypoint correspondences and rigid transformations between two partial point clouds. Our approach differs from these works by analyzing exact and robust recovery conditions of a continuous optimization strategy, and supports non-rigid deformation between the source scan and target.

## 3 Approach

This section presents our technical approach. We begin with a problem statement and an approach overview in Section 3.1. Sections 3.2 through 3.5 elaborate on the technical details.

### 3.1 Problem Statement and Approach Overview

**Problem statement.** Our goal is to establish dense correspondences between a partial scan $S = \{\boldsymbol{p}_1, \cdots, \boldsymbol{p}_n\}$ of $n$ points and a complete template model $\mathcal{M} = (\mathcal{V}, \mathcal{E})$. Here $\mathcal{V} = \{\boldsymbol{q}_1, \cdots, \boldsymbol{q}_N\}$ are the template's $N$ 3D vertices, and $\mathcal{E} \subset \{1, \cdots, N\} \times \{1, \cdots, N\}$ collects pairs of adjacent vertices. Note that the template model is shared by all partial scans. In this paper, we assume the number of points $n$ is shared across all partial scans.

**Approach overview.** The key idea of our approach is to formulate correspondence computation as predicting dense local transformations between $S$ and $\mathcal{M}$, i.e., a rigid transformation $\{R_i \in SO(3), \boldsymbol{t}_i \in \mathbb{R}^3\}$ associated with each point $\boldsymbol{p}_i$ of $S$. The final correspondence of $\boldsymbol{p}_i$ is then given by the nearest neighbor of $R_i\boldsymbol{p}_i + \boldsymbol{t}_i$ on $M$. This representation exhibits appealing flexibility in enforcing the smoothness of correspondences and the articulated structure of body deformation.

As illustrated in Figure 1, our approach combines a transformation initialization module and a transformation regularization module. The initialization module (Section 3.2) first establishes dense correspondences via learned global descriptors (i.e., the output of a descriptor tower). It then fits initial transformations to the resulting correspondences. The initial transformations can have large amounts of noise, both in the form of *local error*—non-smoothness and drift, particularly in flat, featureless areas of the human body—and *structure error* due to ambiguities and symmetry at the global scale (so that i.e. the left hand on the scan maps to the left hand of the template and vice-versa). The transformation regularization module reduces these errors using a synchronization procedure that promotes smoothness (neighboring local transformations are similar) and articulation (local transformations form clusters). We first formulate regularization as a variational problem on the adjacency graph of $\mathcal{S}$ (Section 3.3). We then analyze conditions on the input noise and the structure of the adjacency graph which guarantee that the optimal solution is smooth and recovers articulated

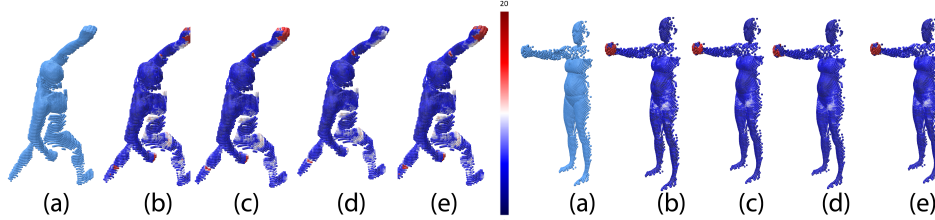

| (a) | (b) | (c) | (d) | (e) | (a) | (b) | (c) | (d) | (e) |

Figure 2: This figure illustrates how correspondences are improved through our pipeline. (a) Input scan. (b-e) show errors (cm) of correspondences derived from learned descriptors, correspondences derived from initial transformation, correspondences derived from the transformation synchronization module, and correspondences derived from the generic transformation synchronization procedure.

deformations. We apply these conditions to the design of a recurrent neural network which solves the variational problem (Section 3.4) and show how to train the entire network end-to-end (Section 3.5).

## 3.2 Initial Transformation Module

This module is composed of a descriptor sub-module for generating dense correspondences and a regression sub-module for fitting initial transformations to these correspondences.

**Descriptor sub-module.** Since we assume a single template model $M$ will be used for all scans, we precompute a dense descriptor on the template model and train a neural network to predict on a scan $S$ the descriptor of the corresponding points on $M$. We leverage Laplacian embedding [5] as our feature descriptor. Specifically, let $L_{N \times N}$ be the discrete cotangent Laplacian matrix on $\mathcal{M}$, and $(\lambda_i, \boldsymbol{u}_i)$ its eigenpairs, sorted in ascending order of eigenvalue magnitude. Let $U_{N \times K}$ be the matrix with columns $\boldsymbol{u}_i/\sqrt{\lambda_i}$ (We use $K = 50$ in this paper); then we assign to vertex $\boldsymbol{q}_l$ of $\mathcal{M}$ the feature descriptor $\overline{\boldsymbol{f}}_l = U^T \boldsymbol{e}_l \in \mathbb{R}^K$. Our motivation for this descriptor is the Euclidean distance in the descriptor space well-approximates diffusion distance on the original mesh [5], an intrinsic distance metric. Therefore, points that are close to each other in the ambient space, but are intrinsically far from each other, naturally separate in descriptor space; moreover, unlike descriptors based on extrinsic geometry, points in smooth regions also remain well-separated in descriptor space. These properties promote spatial consistency in the initial correspondences that we compute.

Given a training set $\mathcal{S} = \{S\}$ of partial scans, for each scan $S$, let $C_S \in \{0, 1\}^{n \times N}$ denote (potentially sparse) ground-truth correspondences between vertices of $S$ and $M$. We train the descriptor sub-module $f_\Theta : \mathbb{R}^{n \times 3} \to \mathbb{R}^{n \times K}$ by minimizing the following regression loss:

$$\mathcal{L}_{des}(\mathcal{S}) := \sum_{S \in \mathcal{S}} \left\| \left[ C_S C_S^T \right] f_\Theta(S) - C_S U \right\|_{\mathcal{F}}^2, \tag{1}$$

where $C_S C_S^T$ masks out the vertices with no ground-truth correspondences. In this paper, we employ PointNet++ [39] for $f_\Theta$, which outperformed GeodesicCNN [36] and image- convolution [51].

**Regression sub-module.** For each point $\boldsymbol{p}_i \in S$ with descriptor $\boldsymbol{f}_i$ predicted by $f_\Theta$, we find its nearest neighbor in descriptor space: $c_i = \arg\min_{1 \leq c \leq N} \|\boldsymbol{f}_i - \overline{\boldsymbol{f}}_c\|^2$. We then compute an initial local transformation $(R_i^{in}, \boldsymbol{t}_i^{in}) \in SE(3)$ for $\boldsymbol{p}_i$ by solving

$$R_i^{in}, \boldsymbol{t}_i^{in} = \underset{R, \boldsymbol{t}}{\operatorname{argmin}} \sum_{j \in \mathcal{N}(i)} \exp\left( - \left\| \boldsymbol{f}_j - \overline{\boldsymbol{f}}_{c_j} \right\|^2 / 2\sigma^2 \right) \left\| R \boldsymbol{p}_j + \boldsymbol{t} - \boldsymbol{q}_{c_j} \right\|^2 \tag{2}$$

where $\mathcal{N}(i)$ collects the $k$-nearest neighbors of $\boldsymbol{p}_i$ on $S$ ($k = 20$ in this paper); $\sigma$ is a trainable hyper-parameter. Equation (2) can be solved in closed form [18] (see supplemental material).

## 3.3 A Generic Transformation Synchronization Formulation

Correspondences derived from initial transformations exhibit both local error and structural errors (See Figure 2(c)). Our ultimate goal is to develop a neural module to rectify these errors. In this section, we will first present transformation regularization in the form of a traditional variational problem, and then we describe how to adjust it to yield our transformation synchronization module.

Specifically, let $\boldsymbol{v}_i^{in} = \left[ \alpha \cdot \text{vec}(R_i^{in}); \boldsymbol{t}_i^{in} \right] \in \mathbb{R}^{12}$ encode the initial local transformation associated with $\boldsymbol{p}_i$; here $\alpha$ is a trainable parameter that weights rotations and translations. We formulate the

following optimization problem to recover regularized transformation vectors:

$$\{\boldsymbol{v}_1^\star, \cdots, \boldsymbol{v}_n^\star\} = \underset{\boldsymbol{v}_i, 1 \leq i \leq n}{\operatorname{argmin}} \sum_{i=1}^n w_i \|\boldsymbol{v}_i - \boldsymbol{v}_i^{in}\| + \sum_{(i,j) \in \mathcal{E}_S} w_{ij} \|\boldsymbol{v}_i - \boldsymbol{v}_j\| \tag{3}$$

where $\mathcal{E}_S$ consists of edges that connect neighboring points in $S$; $\{w_i\}$ and $\{w_{ij}\}$ are weights associated with points and edges in $S$. Note that our formulation (3) is related to network lasso [16]. However, our focus is on analyzing the impact of $\{w_i\}$, $\{w_{ij}\}$, and $\mathcal{E}_S$ on the optimized $\{\boldsymbol{v}_i^\star\}$.

We adopt iteratively reweighted least squares (or IRLS) [9] for optimization. Specifically, we introduce iteratively updated weights $\overline{w}_i^{(s)}$ and $\overline{w}_{ij}^{(s)}$, where $s$ is the iteration count, and modify (3) to

$$\{\boldsymbol{v}_i^{(s)}, 1 \leq i \leq n\} = \underset{\boldsymbol{v}_i, 1 \leq i \leq n}{\operatorname{argmin}} \sum_{i=1}^n w_i \overline{w}_i^{(s)} \|\boldsymbol{v}_i - \boldsymbol{v}_i^{in}\|^2 + \sum_{(i,j) \in \mathcal{E}_S} w_{ij} \overline{w}_{ij}^{(s)} \|\boldsymbol{v}_i - \boldsymbol{v}_j\|^2. \tag{4}$$

We then compute the weights for the next iteration using the formulas

$$\overline{w}_i^{(s+1)} = \delta(\|\boldsymbol{v}_i^s - \boldsymbol{v}_i^{in}\| \leq c_0 c^s), \qquad \overline{w}_{ij}^{(s+1)} = \delta(\|\boldsymbol{v}_i^s - \boldsymbol{v}_j^s\| \leq c_0 c^s) \tag{5}$$

where $c_0$ and $c < 1$ are hyper-parameters. Solving Equation (4) amounts to solving a sparse linear system, which can be done efficiently with Gauss-Seidel iterations,

$$\boldsymbol{v}_i^{(s),(t+1)} \leftarrow \left( w_i \overline{w}_i^{(s)} \boldsymbol{v}_i^{in} + \sum_{j \in \mathcal{N}(i)} w_{ij} \overline{w}_{ij}^{(s)} \boldsymbol{v}_j^{(s),(t)} \right) \Big/ \left( w_i \overline{w}_i^{(s)} + \sum_{j \in \mathcal{N}(i)} w_{ij} \overline{w}_{ij}^{(s)} \right), \tag{6}$$

where $t$ is the inner iteration count. Now we analyze this optimization procedure.

**Noise model.** Let $\boldsymbol{v}_i^{gt}$ be the ground-truth transformations. We assume there are $K$ connected rigidity clusters $\mathcal{V}_k$, with $\boldsymbol{v}_i^{gt} = \boldsymbol{v}_j^{gt}$ whenever $i, j \in \mathcal{V}_k$. We further assume that each cluster can be partitioned into a good vertex set $\mathcal{V}_k^G$ and an outlier set $\mathcal{V}_k^B$ where for each vertex $i$ in the good set, $\boldsymbol{v}_i^{in}$ satisfies $\|\boldsymbol{v}_i^{in} - \boldsymbol{v}_i^{gt}\| \leq \epsilon$. The $\boldsymbol{v}_i^{in}$ for the outlier set are arbitrary.

**Theorem 1.** *Let $D_W = \operatorname{diag}(w_i)$ and $L_W$ be the diagonal matrix that encodes vertex weights and the weighted Laplacian matrix that corresponds to the edge weights, respectively. With $D_W^g$ and $D_W^b$ we collect the rows of $D_W$ that correspond to good and outlier vertices, respectively. Similarly, let $L_W^{g,\mathrm{ol}}$ and $L_W^{b,\mathrm{ol}}$ be the Laplacian matrices that correspond to boundary edges that involve good vertices and outlier vertices, respectively. Suppose there exists a constant $c_1 > 1$ so that*

$$\|(D_W + L_W)^{-\frac{1}{2}}\|_1^2 \cdot \max\left( \|\frac{1}{c_1} D_W^g + 2 L_W^{g,\mathrm{ol}}\|_1, \|D_W^b + 2 L_W^{b,\mathrm{ol}}\|_1 \right) \leq \frac{c}{2+c} \tag{7}$$

*Then the above IRLS procedure recovers vectors $\boldsymbol{v}_i^\star$ that satisfy $\|\boldsymbol{v}_i^\star - \boldsymbol{v}_i^{gt}\| \leq c_1 \epsilon, 1 \leq i \leq n$. Note that $\|A\|_1 = \max\limits_{1 \leq i \leq m} \sum\limits_{j=1}^n |a_{ij}|, \forall A \in \mathbb{R}^{m \times n}$.*

*Proof:* Please refer to the supplemental material. $\square$.

One implication of Theorem 1 is that noisy removal is guaranteed whenever wrong initial transformations connect to many more correct transformations. This suggests that Equation (3) is most effective when points with structural error have many neighbors in $\mathcal{E}_S$, i.e., $\lambda_{\min}(D_{W_k} + L_{W_k})$ is big. As we will discuss shortly, this implication motivates us to introduce down-sampling layers to assess big neighborhoods with similar neighborhood size. Another observation is that Equation (3) becomes more effective, i.e., to make $\|e_k^V\|$ and $\|e_k^E\|$ small, if $w_i$ and $w_{ij}$ are set (using external signals) to be smaller when $\boldsymbol{v}_i^{in}$ is an outlier, and when $\boldsymbol{p}_i$ and $\boldsymbol{p}_j$ fall into different underlying clusters, respectively. We next describe how to our network design, which applies the analysis described above.

## 3.4 Transformation Regularization Module

The regularization module consists of two sub-modules and two network layers derived from the optimization formulation described in the previous section. We first describe each component, then explain how they're stitched together into the regularization module.

**Transformation propagation sub-module.** Given a fixed point cloud $S$, the associated initial transformations, and an adjacency edge set $\mathcal{E}_S$, the transformation propagation sub-module fixes the vertex weights $w_i \overline{w}_i^s$ and the edge weights $w_{ij} \overline{w}_{ij}^s$ and applies four Gauss-Seidel updates (Equation 6) to the transformations $\boldsymbol{v}_i$. It is clear that these updates can be framed as graph convolutions [57, 52], with a specific kernel derived from solving Equation (3).

**Reweighting sub-module.** This sub-module updates the weights associated with points and graph edges $\{\overline{w}_i^{s+1}\}$ and $\{\overline{w}_{ij}^{s+1}\}$. Since truncated weights are hard to learn, we rewrite the update scheme of Equation (5) as

$$\overline{w}_i^{(s+1)} = \epsilon / \sqrt{\epsilon^2 + \left\| \boldsymbol{v}_i^{(s)} - \boldsymbol{v}_i^{in} \right\|^2}, \qquad \overline{w}_{ij}^{(s+1)} = \epsilon / \sqrt{\epsilon^2 + \left\| \boldsymbol{v}_i^{(s)} - \boldsymbol{v}_j^{(s)} \right\|^2}$$

where $\epsilon$ is a trainable parameter. To further increase the adaptivity of the method, we also train the initial weights $w_i$ and $w_{ij}$ to incorporate additional signals. Specifically, let $\boldsymbol{p}_i^c$ be the image of $\boldsymbol{p}_i$ under its current transformation $\boldsymbol{v}_i$. Denote by $c_i$ its closest point on $M$. We set $w_i = \gamma / \sqrt{\gamma^2 + \|\boldsymbol{p}_i^c - \boldsymbol{q}_{c_i}\|^2}$, where $\gamma$ is trainable parameter shared by all points on all scans. Intuitively, if the distance $\|\boldsymbol{p}_i^c - \boldsymbol{q}_{c_i}\|$ is large, then the transformation $\boldsymbol{v}_i$ is unreliable, and so we want to ignore it and instead rely on transformations propagated from $\boldsymbol{p}_i$'s neighbors. To set $w_{ij}$, we evaluate a two-layer fully-connected network that takes the descriptors $\boldsymbol{f}_i$ and $\boldsymbol{f}_j$ of $\boldsymbol{p}_i$ and $\boldsymbol{p}_j$, and current transformations $\boldsymbol{v}_i$ and $\boldsymbol{v}_j$, as input. For the feature descriptor we've chosen based on Laplacian embedding, points that belong to the same rigid part tend to be close in descriptor space [56]), so that we can expect the network to be able to classify whether $\boldsymbol{p}_i$ and $\boldsymbol{p}_j$ belong to the same rigid limb based on their feature vectors.

**Graph down-sampling layer.** This layer is inspired by the max-pooling layer in convolution neural networks. In our setting, it allows us to synchronize transformations over a large spatial neighborhood without directly increasing the number of nearest neighbors. Given an input scan $S_{\text{den}}$, the output of the down-sampling layer is a point cloud $S_{\text{spa}}$ that is randomly sampled from $S_{\text{den}}$, and contains $|S_{\text{spa}}| = |S_{\text{den}}|/2$ points. Let $\mathcal{N}_{\text{den}}(i)$ be the set of 8 nearest neighbors of $\hat{\boldsymbol{p}}_i \in S_{\text{spa}}$ in $S_{\text{den}}$. We set the initial transformation $\hat{\boldsymbol{v}}_i^{in}$ for $\hat{\boldsymbol{p}}_i$ by taking the geometric median of the dense neighbors:

$$\hat{\boldsymbol{v}}_i^{in} = \min_{\boldsymbol{x}} \sum_{j \in \mathcal{N}_{\text{den}}(i)} \|\boldsymbol{x} - \boldsymbol{v}_j\|. \tag{8}$$

We compute this median using reweighted least squares with no trainable parameter.

**Graph up-sampling layer.** The up-sampling layer propagates the synchronization results on a sparse scan $S_{\text{spa}}$ to a dense scan $S_{\text{den}}$ using simple averaging. Let $\mathcal{N}_{\text{spa}}(i) = \{j \,|\, i \in \mathcal{N}_{\text{den}}(j)\}$; we set

$$\boldsymbol{v}_i^{in} = \frac{1}{|\mathcal{N}_{\text{spa}}(i)|} \sum_{j \in \mathcal{N}_{\text{spa}}(i)} \hat{\boldsymbol{v}}_i, \tag{9}$$

where $\hat{\boldsymbol{v}}_i$ is the transformation for point $\hat{\boldsymbol{p}}_i \in S_{\text{spa}}$.

**Network design.** Our regularization module first resolves coarse-scale structural errors on a down-sampled scan, then fine-tunes the local transformations on the original, up-sampled scans. The module consists of two down-sampling layers and two up-sampling layers, with three alternations of transformation propagation and reweighting after each down-sampling or up-sampling layer. Note that the network weights of the reweighting modules are shared.

### 3.5 Network Learning

We define the following transformation loss on the training dataset $\mathcal{S}$, which measures the accuracy with which each point's local transformation explains the deformation of its neighborhood:

$$\mathcal{L}_{trans}(\mathcal{S}) := \sum_{S \in \mathcal{S}} \sum_{(i,c_i) \in \mathcal{C}_S} \sum_{j \in \mathcal{N}(i)} \|R_j \boldsymbol{p}_i + \boldsymbol{t}_j - \boldsymbol{v}_{c_i}\|^2, \tag{10}$$

where $\mathcal{C}_S$ is the set of indices of the nonzero entries in $C_S$. Combining (1) and (10), we arrive at the following term for training the entire network:

$$\min_{\Theta, \Phi} \mathcal{L}_{des}(\mathcal{S}) + \lambda \mathcal{L}_{trans}(\mathcal{S}), \tag{11}$$

where $\lambda$ is optimized on a hold-out validation set.

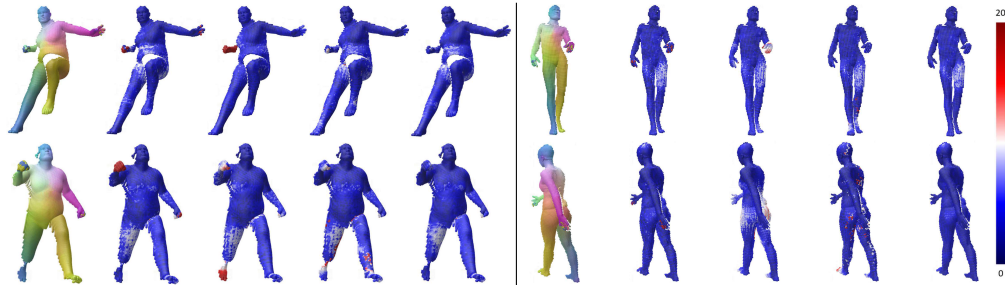

Figure 3: Qualitative comparisons between our approach and top performing baseline. The bottom right is from FAUST while the rest are from SHREC19. For each group, from left to right: predicted correspondences of Ours-Desc, errors (cm) of Ours-Desc, SMPL, ASAP and Ours-Refine.

# 4 Experimental Results

This section presents an experimental evaluation of our proposed approach. We begin by describing the experimental setup in Section 4.1. We then analyze the experimental results in Section 4.2, including baseline comparisons and an ablation study.

## 4.1 Experimental Setup

**Datasets.** We consider three datasets for training and experimental evaluation. The first dataset is **SURREAL** [47], which contains synthetic human animation sequences generated from the SMPL deformation model [33]. We adopt the publicly available[1] 200K pose and shape parameters generated by the authors of [13]. The training/testing split is 190K:10K. The second dataset is **FAUST** [6], which contains 50 inter-subject pairs (FAUST-Inter) and 50 intra-subject pairs (FAUST-Intra). The third dataset is **SHREC19** [34], where the 3D models exhibit certain domain gaps from those of the first two datasets. Note that our network is only trained on SURREAL. FAUST and SHREC19 test its generalizability. To train and evaluate the performance of matching partial scans to complete objects (the *partial-2-full* problem), we follow the protocol of [51] to generate partial scans. Specifically, we render 20 scans for each mesh from SURREAL dataset and 100 scans for FAUST and SHREC19 from random viewpoints. Note that our approach can be easily adapted to compute correspondences between two complete shapes (the *full-2-full* problem) by synchronizing partial-2-full matching (c.f. [51]). Due to the space constraint, we leave these evaluations to the supplemental material.

**Baseline comparisons.** We consider three categories of approaches for baseline comparison. The first category replaces the spectral descriptors with three invariant signatures, namely, HKS [45], WKS [4] and SHOT [43]. They evaluate the importance of using the Laplacian embedding descriptor in our framework. Many partial-2-full matching are based on deforming the template. The second category of baselines consists of two template deformation approaches, i.e., ASAP [19] and SMPL [33]. Since these approaches require initial correspondences, we combine them with the dense correspondences computed by our descriptor tower (shown as Ours-Desc). Such comparisons directly assess the performance of our transformation regularization module. The third category is DHBC [51], which is a state-of-the-art dense correspondence approach based on deep learning. Many prior approaches operate in the full-2-full matching setting. In the supplemental material, we compare against four state-of-the-art approaches, namely, DeepGeoFunc [11], DHBC [51], 3D-CODED [13], and LES [10].

**Evaluation protocol.** We employ the evaluation protocols of [51] for evaluating computed correspondence results. We report the average correspondence error and the recall rate within 5cm and 10cm. The unit of correspondence error is centimeter (cm). The statistics are shown in Table 1.

## 4.2 Analysis of Results

Table 1 and Figure 3 present quantitative and qualitative results, respectively. Overall, our approach yielded highly accurate results. On SURREAL, FAUST, and SHREC19, our approach achieved 1.71cm, 1.90cm, and 4.81cm mean errors, respectively. Moreover, the 10cm recall rates on these datasets are 99.2%, 99.8%, and 97.0%, respectively. Note that the results on SHREC19 are much worse than SURREAL and FAUST. This is due to the domain gap between SURREAL and SHREC19.

| Method | SURREAL | | | FAUST | | | SHREC19 | | |
|---|---|---|---|---|---|---|---|---|---|
| | AE(cm) | 5cm-recall | 10cm-recall | AE(cm) | 5cm-recall | 10cm-recall | AE(cm) | 5cm-recall | 10cm-recall |
| SHOT | 5.97 | 0.803 | 0.942 | 6.66 | 0.665 | 0.893 | 11.04 | 0.477 | 0.840 |
| HKS | 7.70 | 0.719 | 0.933 | 9.98 | 0.631 | 0.861 | 11.28 | 0.462 | 0.829 |
| WKS | 8.99 | 0.676 | 0.869 | 10.15 | 0.622 | 0.859 | 11.25 | 0.467 | 0.831 |
| DHBC | 14.29 | 0.440 | 0.651 | 10.91 | 0.503 | 0.772 | 17.24 | 0.401 | 0.646 |
| Ours-Desc | 2.19 | 0.939 | 0.991 | 2.59 | 0.917 | 0.994 | 5.84 | 0.749 | 0.965 |
| Ours-Opt | 2.13 | 0.942 | 0.991 | 2.44 | 0.919 | 0.995 | 5.77 | 0.752 | 0.966 |
| SMPL | 2.02 | 0.937 | 0.977 | 1.98 | 0.932 | 0.973 | 5.48 | 0.751 | 0.897 |
| ASAP | 1.93 | 0.949 | 0.981 | 1.95 | 0.946 | 0.983 | 6.15 | 0.774 | 0.923 |
| Ours-Refine | **1.71** | **0.960** | **0.992** | **1.90** | **0.953** | **0.998** | **4.81** | **0.810** | **0.970** |

Table 1: Evaluation of all correspondence computation methods for partial-2-full correspondence task. We report average correspondence error, 5cm-recall and 10cm-recall of all correspondence computation methods on each dataset. Ours-Refine represents the result of our method after transformation regularization.

**Baseline comparisons on dense point-wise descriptors.** Table 1 shows baseline comparisons where we replace the Laplacian descriptor with three baseline shape descriptors, i.e., SHOT [43], HKS [45] and WKS [4]. The mean error of the top-performing baseline increased by 173%, 204%, and 89% on SURREAL, FAUST and SHREC19, respectively. Accordingly, the 10cm recall rates dropped from 99.1%, 99.4%, and 96.5% to 94.2%, 89.3%, and 84.0%, respectively. These performance drops are due to the fact that although these baseline signatures are distinctive for extremal points, they do not provide sufficient separations between points among smooth regions. As a result, there are large errors in initial local transformations, which affect the final performance.

**Baseline comparisons on regularization.** Table 1 shows baseline comparisons with ASAP [19] and SMPL [33]. We can see that our approach outperforms these two baseline approaches considerably. The mean errors of SMPL/ASAP on SURREAL, FAUST, and SHREC19 are 2.02cm/1.93cm, 1.98cm/1.95cm, and 5.48cm/6.15cm, respectively. These statistics show that our regularization module, which leverages a trainable recurrent module to suppress diverse types of errors in the initial transformations, is superior to the generic regularization strategies in ASAP and SMPL.

**Comparison with [51].** Our approach reduces the mean errors of [51] by 88.8%, 82.8%, and 72.1% on SURREAL, FAUST and SHREC19, respectively, which are quite salient. The improvements come from both the Laplacian-based descriptors and our regularization module.

**Ablation study.** We present an ablation study on our transformation synchronization module. As shown in Table 1, merely using the initial transformations results in an increase in mean errors by 28.1%, 36.3%, and 21.4% on SURREAL, FAUST and SHREC19, respectively. In other words, the transformation synchronization module, which enforces approximate piece-wise rigidity of the local transformations, is critical for correspondence computation. Note that initial transformations are obtained through local robust regression, which already has certain denoising power. Specifically, the mean errors of nearest neighbors in the descriptor space are bigger than those of the initial transformations by 2.82%, 2.70%, and 5.31% on these three datasets, respectively. We also analyzed the performance of using the generic convex optimization strategy (shown as Ours-Opt in Table 1) in Section 3.3. Although we can observe certain reductions in mean errors, e.g., 2.73%, 5.79%, and 1.20%, the reductions are not as salient as the learned regularization module.

**Identification of rigid parts.** To further understand our approach's performance, we can cluster the final optimized local transformations to see whether the resulting clusters align with semantic parts of a human model. To this end, we apply mean-shift clustering on the final local transformations. We then visualize the cluster structures on the input scan. We can see that the resulting clusters are consistent with the main parts of a human subject, which indicates that our approach automatically recovers the underlying articulated deformations. Please refer to the supp. material for more results. 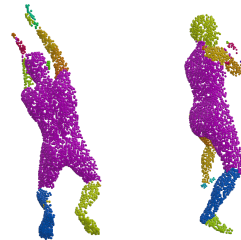

**Full-2-full baseline comparisons.** As detailed in the supplemental material, our approach compares favorably against all four baselines for full-2-full shape matching. These statistics again show the advantages of using local transformations as the data representation for regularizing dense correspondences when compared to other data representations.

# 5 Conclusions

In this paper, we integrated a new representation into an end-to-end trainable neural network and proposed a transformation regularization module that instills articulated motion structure. Previous works are either enforcing predicted correspondences to be smooth, or to preserve geodesic distances which are hard to compute on partial scans. Our approach builds upon an extrinsic deformation structure, which gives us a huge advantage in network design and training. Note that any dense descriptor module tends to work well on areas that possess rich features, while transformation regularization module tends to propagate good transformations among smooth regions. The complementary nature of these two modules are critical to our approach.

# 6 Statement of Broader Impacts

Computing dense correspondences between a partial scan and a template, or between two partial scans, is a fundamental task for analyzing and understanding 3D data captured from the real world. Our work is foundational, improving the accuracy and robustness of this important task, and will benefit downstream applications that rely on the ability to find accurate dense correspondences.

One such application area is human subject tracking, where the correspondences between the partial scan the the complete template model can be used to deform the template to obtain complete deformed shape that corresponds to each partial scan. Our research will allow reconstruction of higher-fidelity animation sequences that better captures nuanced motion from large-scale, real-world data. Applications that will benefit from this improved tracking include imitation learning, where a system can learn from motion of each observed subject, especially of fine motor skills not able to be tracked before; movie/game industry, where one can insert the reconstructed motion of an actor into virtual environment, with unprecedented expressiveness of the reconstructed actor; and sports, where one can reconstruct and analyze the athletes' motions to make recommendations both for improving athletic performance as well as enhancing athlete safety.

Another application area is full body reconstruction from a few scans. In this setting, the template mesh serves as an intermediate object to establish dense correspondences between partial scans. Our research represents an important steps towards allowing ordinary users to scan themselves with high accuracy at home using commodity hardware. Access to a high-quality digital avatar facilitates many applications such as virtual fitting for on-line shopping, improved telepresence and telemedicine, and new forms of entertainment and social media where users can place and animate themselves in a 3D environment.

Potential abuses and negative impacts of improved tracking and reconstruction include the ability to identify people without their consent, based on body shape or motion characteristics, in settings where traditional facial recognition algorithms fail. 3D avatars of a person reconstructed from surreptitious partial scans might also be used to create "deep fakes" or to otherwise infringe on the privacy rights of the subject.

From a technical perspective, the problem falls into the category of structure prediction that combines point-wise predictions and priors on correlations among multiple points. Unlike the standard MRF formulation, this paper explores a new data representation, which turns structure prediction into a continuous optimization problem. This methodology can inspire future research on relevant problems, where the problem space lies in a continuous domain. Moreover, there is growing interest in turning optimization problems into neural networks with hyper-parameters trained end-to-end. Our approach contributes to this effort, and we hope the insights we used to design the resulting neural network for training (including our analysis of robust recovery conditions for the transformation synchronization problem) can be applied to and stimulate future research on similar problems.

Finally, like any algorithm for computing dense correspondences, our approach is not *guaranteed* to generate correct correspondences in all the settings. Additional checks and verification (by humans using interactive tools, for instance) should be used to validate and rectify the outputs, especially if the results are used in safety- or health-critical applications such as personalized medicine.

# 7 Acknowledgement

We thank the anonymous reviewers for their valuable comments. Qixing Huang would like to acknowledge support from NSF DMS-1700234, NSF CIP-1729486, NSF IIS-1618648, NSF HDR TRIPODS-1934932, a gift from Snap Research and a GPU donation from Nvidia Inc.

## Footnotes

[1] `https://github.com/ThibaultGROUEIX/3D-CODED/tree/master/data`

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
