[Supplementary Material]

# A  Overview of Supplemental Material

We organize supplementary material as follows. In Section B, we present our results compared to baselines on the *full-to-full* matching task between two complete shapes. In Section C, we show a small set of qualitative results for animal shapes. In Section D, we explain the implementation details of reweighting module, which is an important component of our transformation regularization module. In Section E, we detail the back-propagation rule for the regression submodule that estimates local transformations, as described in Equation (2). Finally, in Section F we prove Theorem 1 on the exact recovery conditions for our transformation sychronization formulation.

# B  Full-to-full Matching Results

In this section, we compare our method with the four baselines, namely, DeepGeoFunc [11], DHBC [51], 3D-CODED [13], and three variants of LES [10]: point translation and patch deformation 3D (LES-PTD3), point translation 10D (LES-PT10), patch deformation 10D (LES-PD10). All correspondences are refined by designated algorithms: 3D-CODED and LES use deformation during prediction, DeepGeoFunc uses ZoomOut [35], and DHBC and our method uses non-rigid ICP for correspondence refinement. All statistics are reported in Table 2 and qualitative results are provided in Figure 4.

Overall, our method achieved competitive results when compared against these state-of-the-art baselines. Specifically, we achieved 1.74cm/1.72cm/1.42cm/3.50cm average correspondence error on the SURREAL/FAUST-inter/FAUST-intra/SHREC19 datasets, where the top-performing baseline achieved 2.54cm/1.64cmm/1.45cm/5.02cm. We achieved the most improvement on SHREC19, which is recognized as the most difficult dataset due to large domain gap. This result demonstrates the power of our regularization module.

It should be noted that all the baseline approaches involve isometry-invariance constraints (e.g., spectral techniques and those preserve geodesic distances) possess strong regularizations in the full-2-full setting. Still, our approach, which leverages the implicit articulated deformation structure, outperform them by a considerable margin. This shows the advantages of modeling approximated piece-wise rigidity constraints for establishing dense correspondences between 3D models of humans.

| Method | SURREAL | | | FAUST-inter | | | FAUST-intra | | | SHREC19 | | |
|---|---|---|---|---|---|---|---|---|---|---|---|---|
| | AE(cm) | 5cm-recall | 10cm-recall | AE(cm) | 5cm-recall | 10cm-recall | AE(cm) | 5cm-recall | 10cm-recall | AE(cm) | 5cm-recall | 10cm-recall |
| DHBC | / | / | / | 2.35 | 0.900 | 0.972 | 2.00 | 0.911 | 0.975 | / | / | / |
| 3D-CODED | 2.89 | 0.958 | 0.971 | 2.08 | 0.956 | 0.983 | 1.97 | 0.964 | 0.986 | 6.69 | 0.740 | 0.898 |
| LES-PTD3 | 2.74 | 0.962 | 0.971 | 1.64 | 0.969 | 0.992 | 1.49 | 0.976 | 0.992 | 6.75 | 0.742 | 0.894 |
| LES-PT10 | 2.54 | 0.962 | 0.971 | 1.69 | 0.971 | 0.991 | 1.45 | 0.976 | 0.992 | 6.95 | 0.733 | 0.895 |
| LES-PD10 | 2.75 | 0.962 | 0.972 | 2.05 | 0.949 | 0.978 | 2.32 | 0.949 | 0.971 | 5.02 | 0.766 | 0.948 |
| DeepGeoFunc | 3.45 | 0.905 | 0.971 | 2.92 | 0.828 | 0.972 | 2.11 | 0.886 | 0.975 | 17.02 | 0.440 | 0.727 |
| Ours-Refine-ICP | 1.74 | 0.980 | 0.990 | 1.72 | 0.962 | 0.990 | 1.42 | 0.969 | 0.992 | 3.50 | 0.824 | 0.968 |

Table 2: Evaluation of all correspondence computation methods for the full-to-full correspondence task. We report the average correspondence error, 5cm-recall, and 10cm-recall for all correspondence computation methods on each dataset. The unit of error is cm. All reported correspondences results are evaluated after designated refinement methods have been applied. All the results are reported the same set of datasets reported in this paper by running the code from the authors of the baseline approaches.

# C  Qualitative Results on Animal Shapes

As an additional experiment, we train our model on a synthetic dataset utilizing SMAL [61]. We generate 20K triangular meshes using latent parameters generated from a gaussian distribution. For each mesh, we rendered 100 depth scans from random viewpoints. The model was trained in a similar fashion as the human shape and evaluated on the horse class of TOSCA dataset [7] (see Figure 5). The results indicates the usefulness of our transformation synchronization module. For more results, please refer to our code base.

# D  Implementation of Reweighting Module

In this section, we demonstrate the implementation details of reweighting module.

Figure 4: Qualitative comparisons between our approach and top performing baseline on FAUST-inter, FAUST-intra and SHREC19. For each dataset, we report from left to right: target mesh, error maps of two top-performing baselines and our method after refinement. For each method, the error (cm) is visualized on the source mesh.

Figure 5: Qualitative Results evaluated on the horse class of TOSCA dataset [7]. From left to right: 1) Ground Truth Correspondence on source shape. 2) Initial Correspondence before transformation regularization. 3) Correspondence after transformation regularization.

The input to this module is a point cloud with vertex-wise correspondences and transformations. The goal is to update vertex weights and edge weights to prepare for transformation propagation by (6).

The design of reweighting module is illustrated in figure 6. Specifically, we propagate messages $\boldsymbol{m}_i^{(k)}$ among vertices and aggregate them by element-wise maximum $MAX(\cdot)$ as in the message passing framework [12]. This is repeated by $k_{\max}$ iterations, where in the last layer $\boldsymbol{m}_i^{(k_{\max})}$ is a scalar indicating the vertex-wise weights.

Input Point Cloud: $\boldsymbol{x}_i$

Initial Correspondence on template: $\boldsymbol{y}_i$

Initial Transformations: $R_i, \boldsymbol{t}_i$

Initial Embedding: $m_i^{(0)} = \mathrm{MLP}(\boldsymbol{x}_i)$

$$
\boxed{\begin{array}{cc} \boldsymbol{x}_i & \boldsymbol{y}_i \\ m_i^{(0)} & R_i, \boldsymbol{t}_i \end{array}} \rightarrow \boxed{\begin{array}{c} w_{ij} = \sigma(\alpha\|R_i\boldsymbol{x}_j + \boldsymbol{t}_i - \boldsymbol{y}_j\| + \beta) \\ \boldsymbol{m}_i^{(k+1)} = \max_{j\in\mathcal{N}(i)} w_{ij}\boldsymbol{m}_i^{(k)} \end{array}} \rightarrow w_i = \sigma(m_i^{(k_{\max})}) \in \mathbb{R}
$$

Figure 6: Illustration of Reweighting Module. Note that $\boldsymbol{y}_i \in \mathbb{R}^3$ are point locations on template mesh.

# E   Back-Propagation Rule for (2)

We consider a general objective function for pose regression:

$$
R_i^{in}, \boldsymbol{t}_i^{in} = \underset{R,\boldsymbol{t}}{\operatorname{argmin}} \sum_{j\in\mathcal{N}(i)} w_j\|R\boldsymbol{p}_j + \boldsymbol{t} - \boldsymbol{q}_{c_j}\|^2 \tag{12}
$$

where $w_j > 0$. Our goal is derive the partial derivatives of $R_i^{in}$ and $\boldsymbol{t}_i^{in}$ with respect to $w_j$.

The following proposition, which is due to [18], characterizes a closed-form expression for $R_i^{in}$ and $\boldsymbol{t}_i^{in}$.

**Proposition 1.** *Let*

$$
\boldsymbol{c}_p = \sum_{j\in\mathcal{N}(i)} w_j\boldsymbol{p}_j / \sum_{j\in\mathcal{N}(i)} w_j, \qquad \boldsymbol{c}_q = \sum_{j\in\mathcal{N}(i)} w_j\boldsymbol{q}_{c_j} / \sum_{j\in\mathcal{N}(i)} w_j.
$$

*Denote*

$$
S = \sum_{j\in\mathcal{N}(i)} w_j(\boldsymbol{p}_j - \boldsymbol{c}_p)(\boldsymbol{q}_{c_j} - \boldsymbol{c}_q)^T.
$$

*Let $S = U\Sigma V^T$ be the singular-value decomposition of $S$. Then*

$$
R_i^{in} = V \begin{pmatrix} 1 & 0 & 0 \\ 0 & 1 & 0 \\ 0 & 0 & \operatorname{sign}(\det(S)) \end{pmatrix} U^T, \qquad \boldsymbol{t}_i^{in} = \boldsymbol{c}_q - R\boldsymbol{c}_p. \tag{13}
$$

The following proposition characterizes the derivatives of $R_i^{in}$ with respect to elements of $S$, which then determines the derivatives with respect to $w_j$.

**Lemma 1.** *The derivatives of $R$ with respect to $S$ is given by*

$$
dR = \sum_{1\le i\ne j\le 3} \boldsymbol{v}_i \cdot \boldsymbol{u}_j^T \cdot \frac{\boldsymbol{u}_j^T \cdot dS \cdot \boldsymbol{v}_i - \boldsymbol{u}_i^T \cdot dS \cdot \boldsymbol{v}_j}{\sigma_i + \sigma_j}
$$

$$
+ (\operatorname{sign}(\det(S)) - 1) \sum_{j=1}^{2} \left( \boldsymbol{u}_j^T dS\boldsymbol{v}_3 \cdot \frac{\sigma_3 \boldsymbol{v}_3 \boldsymbol{u}_j^T + \sigma_j \boldsymbol{v}_j \boldsymbol{u}_3^T}{\sigma_3^2 - \sigma_j^2} + \boldsymbol{u}_3^T dS\boldsymbol{v}_j \cdot \frac{\sigma_3 \boldsymbol{v}_j \boldsymbol{u}_3^T + \sigma_j \boldsymbol{v}_3 \boldsymbol{u}_j^T}{\sigma_3^2 - \sigma_j^2} \right) \tag{14}
$$

*where $U = (\boldsymbol{u}_1, \boldsymbol{u}_2, \boldsymbol{u}_3)$, $V = (\boldsymbol{v}_1, \boldsymbol{v}_2, \boldsymbol{v}_3)$, and $\Sigma = \operatorname{diag}(\sigma_1, \sigma_2, \sigma_3)$.*

*Proof.* It is clear that

$$
S\boldsymbol{v}_i = \sigma_i\boldsymbol{u}_i, \boldsymbol{u}_i^T S = \sigma_i\boldsymbol{v}_i^T, \qquad 1 \le i \le 3.
$$

Taking the differential on both sides, we obtain

$$
dS \cdot \boldsymbol{v}_i + S \cdot d\boldsymbol{v}_i = d\sigma_i \cdot \boldsymbol{u}_i + \sigma_i \cdot d\boldsymbol{u}_i \tag{15}
$$

$$
d\boldsymbol{u}_i^T S + \boldsymbol{u}_i^T dS = d\sigma_i \boldsymbol{v}_i^T + \sigma_i d\boldsymbol{v}_i^T \tag{16}
$$

Left multiplying both sides of (15) by $\boldsymbol{u}_j$ with $j \neq i$ and observing that $\boldsymbol{u}_j^T \boldsymbol{u}_i = 0$, we obtain

$$\boldsymbol{u}_j^T dS \boldsymbol{v}_i + \boldsymbol{u}_j^T S d\boldsymbol{v}_i = \sigma_i \boldsymbol{u}_j^T d\boldsymbol{u}_i. \tag{17}$$

Similarly right multiplying both sides of (16) by $\boldsymbol{v}_j^T$ with $j \neq i$ gives

$$d\boldsymbol{u}_i^T S \boldsymbol{v}_j + \boldsymbol{u}_i^T dS \boldsymbol{v}_j = \sigma_i d\boldsymbol{v}_i^T \boldsymbol{v}_j. \tag{18}$$

It follows that

$$\boldsymbol{u}_j^T dS \boldsymbol{v}_i + \sigma_j \boldsymbol{v}_j^T d\boldsymbol{v}_i = \sigma_i \boldsymbol{u}_j^T d\boldsymbol{u}_i \tag{19}$$

$$\sigma_j \cdot d\boldsymbol{u}_i^T \boldsymbol{u}_j + \boldsymbol{u}_i^T dS \boldsymbol{v}_j = \sigma_i d\boldsymbol{v}_i^T \boldsymbol{v}_j \tag{20}$$

Observe that $\boldsymbol{u}_j T d\boldsymbol{u}_i + d\boldsymbol{u}_i^T \boldsymbol{u}_j = 0$ and $\boldsymbol{v}_j^T d\boldsymbol{v}_i + d\boldsymbol{v}_i^T \boldsymbol{v}_j = 0$. Combining (19) and (20) to solve for $\boldsymbol{u}_j^T d\boldsymbol{u}_i$ and $\boldsymbol{v}_j^T d\boldsymbol{v}_i$, we obtain

$$\boldsymbol{u}_j^T d\boldsymbol{u}_i = \frac{\sigma_i \boldsymbol{u}_j^T dS \boldsymbol{v}_i + \sigma_j \boldsymbol{u}_i^T dS \boldsymbol{v}_j}{\sigma_i^2 - \sigma_j^2} \tag{21}$$

$$\boldsymbol{v}_k^T d\boldsymbol{v}_i = \frac{\sigma_i \boldsymbol{u}_i^T dS \boldsymbol{v}_j + \sigma_j \boldsymbol{u}_j^T dS \boldsymbol{v}_i}{\sigma_i^2 - \sigma_j^2} \tag{22}$$

Since $\boldsymbol{u}_i d\boldsymbol{u}_i = 0$, we have

$$d\boldsymbol{u}_i = \sum_{j \neq i} \frac{\sigma_i \boldsymbol{u}_j^T dS \boldsymbol{v}_i + \sigma_j \boldsymbol{u}_i^T dS \boldsymbol{v}_j}{\sigma_i^2 - \sigma_j^2} \boldsymbol{u}_j$$

$$d\boldsymbol{v}_i = \sum_{j \neq i} \frac{\sigma_i \boldsymbol{u}_i^T dS \boldsymbol{v}_j + \sigma_j \boldsymbol{u}_j^T dS \boldsymbol{v}_i}{\sigma_i^2 - \sigma_j^2} \boldsymbol{v}_j$$

In the case $\det(S) > 0$, we have

$$dR = \sum \left( \boldsymbol{v}_i d\boldsymbol{u}_i^T + d\boldsymbol{v}_i \boldsymbol{u}_i^T \right)$$

$$\sum_{1 \leq i \neq j \leq 3} \boldsymbol{v}_i \cdot \boldsymbol{u}_j^T \cdot \frac{\boldsymbol{u}_j^T \cdot dS \cdot \boldsymbol{v}_i - \boldsymbol{u}_i^T \cdot dS \cdot \boldsymbol{v}_j}{\sigma_i + \sigma_j}.$$

The proof for the case $\det(S) < 0$ is similar and it omitted for brevity. $\qquad\square$

For the network training, at each iteration and for each instance, we fix the nearest neighbors and apply the expression described above to back-propagation the gradients from the initial transformations to the descriptor tower.

# F   Proof of Robust Recovery Condition

This section presents the proof of Theorem. We begin with a few useful lemmas in Section F.1. We then complete the proof in Section F.2.

## F.1   Useful Lemmas

The first lemma concerns the relation between different norms of a symmetric matrix $A$.

**Lemma 2.** *Given a symmetric matrix A, we have*

$$\|A\| \leq \|A\|_1. \tag{23}$$

*Proof.* Consider any eigenvalue $\lambda$ of $A$. Denote $\boldsymbol{x} = (x_1, \cdots, x_n)^T \in \mathbb{R}^n$ as its corresponding eigenvector. Let

$$i^\star = \operatorname*{argmax}_{1 \leq i \leq n} |x_i|.$$

Then
$$\sum_{j=1}^{n} a_{ij} x_j = \lambda x_{i^\star}.$$

It follows that
$$|\lambda| |x_{i^\star}| \leq \left| \sum_{j=1}^{n} a_{ij} x_j \right| \leq \sum_{j=1}^{n} |a_{ij}| |x_j| \leq \sum_{j=1}^{n} |a_{ij}| |x_{i^\star}| \leq \|A\|_1 \cdot |x_{i^\star}|$$

It follows that
$$|\lambda| \leq \|A\|_1,$$

which ends the proof. □

The second lemma concerns the $L_1$ norm of $(A - B)^{-1}$.

**Lemma 3.** *Suppose* $\|A^{-\frac{1}{2}}\|_1^2 \|B\|_1 < 1$. *Then*

$$\|(A - B)^{-1}\|_1 \leq \frac{\|A^{-\frac{1}{2}}\|_1^2}{1 - \|A^{-\frac{1}{2}}\|_1^2 \|B\|_1}. \tag{24}$$

*Proof.* First of all, since $\|A^{-\frac{1}{2}}\|_1^2 \|B\|_1 < 1$. It follows that
$$\|A^{-\frac{1}{2}} B A^{-\frac{1}{2}}\| < 1.$$

In other words, we have
$$(A - B)^{-1} = A^{-\frac{1}{2}} \left( I - A^{-\frac{1}{2}} B A^{-\frac{1}{2}} \right) A^{-\frac{1}{2}}$$
$$= A^{-\frac{1}{2}} \cdot \sum_{i=0}^{\infty} \left( A^{-\frac{1}{2}} B A^{-\frac{1}{2}} \right)^i \cdot A^{-\frac{1}{2}}.$$

Therefore
$$\|(A - B)^{-1}\|_1 \leq \|A^{-\frac{1}{2}}\|_1^2 \cdot \sum_{i=0}^{\infty} \left\| \left( A^{-\frac{1}{2}} B A^{-\frac{1}{2}} \right)^i \right\|_1$$
$$\leq \|A^{-\frac{1}{2}}\|_1^2 \cdot \sum_{i=0}^{\infty} \left( \|A^{-\frac{1}{2}}\|_1^2 \|B\|_1 \right)^i$$
$$\leq \frac{\|A^{-\frac{1}{2}}\|_1^2}{1 - \|A^{-\frac{1}{2}}\|_1^2 \|B\|_1}.$$

□

### F.2 Completing the Proof

Note that our reweighted least square formulation solves the following linear system at each iteration:

$$\min_{\boldsymbol{v}_i, 1 \leq i \leq n} \sum_{i=1}^{n} w_i^{(t)} \|\boldsymbol{v}_i - \boldsymbol{v}_i^{in}\|^2 + \lambda \sum_{(i,j) \in \mathcal{E}} w_{ij}^{(t)} \|\boldsymbol{v}_i - \boldsymbol{v}_j\|^2 \tag{25}$$

where $w_i^{(t)}, 1 \leq i \leq n$ and $w_{ij}^{(t)}, (i,j) \in \mathcal{E}$ are weights associated with vertices and edges at iteration $t$. Define
$$W_{\mathcal{V}}^{(t)} := \mathrm{Diag}(w_i^{(t)}|_{1 \leq i \leq n}) \in \mathbb{R}^{n \times n}, \quad W_{\mathcal{E}}^{(t)} := \mathrm{Diag}(w_{ij}^{(t)}|_{(i,j) \in \mathcal{E}}) \in \mathbb{R}^{|\mathcal{E}| \times |\mathcal{E}|}.$$

**Proposition 2.** *Suppose* $\boldsymbol{v}_i = \boldsymbol{v}_i^{gt} + \delta_i, 1 \leq i \leq n$ *and* $\delta_{ij} = \boldsymbol{v}_i^{gt} - \boldsymbol{v}_j^{gt}, (i,j) \in \mathcal{E}$. *Then the solution* $\boldsymbol{x}^{(t)}$ *to (25) satisfies*
$$\boldsymbol{v}^{(t)} - \boldsymbol{v}^{gt} := L^{(t)-1} \boldsymbol{r}^{(t)}.$$

*where*

$$L^{(t)} := W_{\mathcal{V}}^{(t)} + \lambda J^T W_{\mathcal{E}}^{(t)} J \tag{26}$$

$$\boldsymbol{r}^{(t)} := W_{\mathcal{V}}^{(t)}\mathrm{mat}(\delta_i) + \lambda J^T W_{\mathcal{E}}^{(t)}\mathrm{mat}(\delta_{ij}). \tag{27}$$

*where $J \in \mathbb{R}^{|\mathcal{E}|\times n}$ is the vertex and edge indicator matrix; $\mathrm{mat}(\delta_i) \in \mathbb{R}^{n\times 12}$ collects the vertex indicators; $\mathrm{mat}(\delta_{ij}) \in \mathbb{R}^{|\mathcal{E}|\times 12}$ collects the edge indicators.*

*Proof.* Let $V = (\boldsymbol{v}_1, \cdots, \boldsymbol{v}_n)^T \in \mathbb{R}^{n\times 12}$. It is clear that (25) is identical to

$$\min_V \ \mathrm{Trace}\big(V^T L^{(t)} V\big) - 2\mathrm{Trace}\big(V^T \boldsymbol{r}^{(t)}\big)$$

which ends the proof. □

Let $\mathcal{V}^{(t)} \subset [n]$ and $\mathcal{E}^{(t)} \subset \mathcal{E}$ be the remaining edges at iteration $t$. By the default setting, we assume $\mathcal{V}^{(t)}$ and $\mathcal{E}^{(t)}$ satisfy the recursion properties:

$$\mathcal{V}^g \subset \mathcal{V}^{(t)} \subset [n]$$
$$\mathcal{E}^g \subset \mathcal{E}^{(t)} \subset \mathcal{E}$$

Moreover,

$$\|\delta_i\| \leq c_0 c^t, \quad \forall 1 \leq i \leq n$$
$$\|\delta_{ij}\| \leq c_0 c^t, \quad \forall (i,j) \in \mathcal{E}$$

It follows that

$$\|\boldsymbol{r}^{(t)}\|_\infty \leq \max\Big( \epsilon_0 \|D_W^g\|_1 + c_0 c^t \|L_W^{g,\mathrm{ol}}\|_1, c_0 c^t (\|D_W^b\|_1 + \|L_W^{b,\mathrm{ol}}\|_1) \Big) \tag{28}$$

Note that the solution at the next iteration satisfies

$$\|V^{(t+1)} - V^{gt}\|_\infty \leq \|{L^{(t)}}^{-1}\|_1 \|\boldsymbol{r}^{(t)}\|_\infty. \tag{29}$$

Applying Lemma 3, we have

$$\begin{aligned}\|{L^{(t)}}^{-1}\|_1 &\leq \frac{\|L^{-\frac{1}{2}}\|_1^2}{1 - \|L^{-\frac{1}{2}}\|_1^2 \|L_{\mathrm{ol}}\|_1} \\ &\leq \frac{\|L^{-\frac{1}{2}}\|_1^2}{1 - \|L^{-\frac{1}{2}}\|_1^2 \max\Big(\|L_W^{g,\mathrm{ol}}\|_1, \|D_W^b\|_1 + \|L_W^{b,\mathrm{ol}}\|_1\Big)}. \end{aligned} \tag{30}$$

where $L = D_W + L_W$, and $L_{\mathrm{ol}}$ collects all the elements of $L$ that are attached to outlier observations. Combing (28)-(30), we have

$$\|V^{(t+1)} - V^{gt}\|_\infty \leq \frac{\|L^{-\frac{1}{2}}\|_1^2 \cdot \max\Big( \epsilon_0 \|D_W^g\|_1 + c_0 c^t \|L_W^{g,\mathrm{ol}}\|_1, c_0 c^t (\|D_W^b\|_1 + \|L_W^{b,\mathrm{ol}}\|_1) \Big)}{1 - \|L^{-\frac{1}{2}}\|_1^2 \max\Big(\|L_W^{g,\mathrm{ol}}\|_1, \|D_W^b\|_1 + \|L_W^{b,\mathrm{ol}}\|_1\Big)} \tag{31}$$

Therefore, under the assumptions of the theorem, i.e.,

$$\|(D_W + L_W)^{-\frac{1}{2}}\|_1^2 \cdot \max\Big(\|\frac{1}{c_1}D_W^g + 2L_W^{g,\mathrm{ol}}\|_1, \|D_W^b + 2L_W^{b,\mathrm{ol}}\|_1\Big) \leq \frac{c}{2+c}$$

we have

$$\|\boldsymbol{r}^{(t)}\|_\infty \leq \frac{1}{2}\max(c_1 \epsilon_0, c_0 c^{t+1})$$

This means the iterative procedure only prunes the outliers, and now we end the proof.