[Reviews · NeurIPS 2020]

Review 1

Summary and Contributions: The authors present a learning algorithm that computes dense correspondences between full and partial 3D human body shapes. Initial correspondences are extracted from a partial input shape via a point cloud feature extractor and matched to the Laplacian embedding of a human template shape via a 6-dim. rigid parameterization per point. The noisy initial matches are then refined using a transformation synchronization module and the whole network is trained end-to-end.

Strengths: - Overall, the quantitative results show a convincing improvement over the baselines (see however my concerns under weaknesses), in particular in combination with the results from Table 2 in the supplementary. - The L1 regularized refinement module and the corresponding IRLS algorithm seem to be a well conceived choice for human shapes with parts of piecewise constant rotations. - In terms of time complexity, the method is extremely efficient in comparison with related work. Overall, I am inclined to give this work a positive initial rating.

Weaknesses: - Regarding the comparisons in Table 1., I would have liked to see comparisons with [9], [10] and/or [11]. SMPL and ASAP are good, but they are blended with an initialization from the authors' method and a comparison with DHBC alone is not completely convincing. - The method requires the ground truth correspondences as supervision which is a strong assumption. Many related methods ([11, 13, 40] in the paper) investigated the unsupervised setting at least to some degree and therefore allow for training on non-synthetic data. - The authors decided to limit themselves to human shapes in this work, although there are no modelling assumption that clearly require this. While I can to some extent understand this decision, it would still be interesting to see some qualitative generalization results to other classes of shapes. I understand that your method requires the human template for training, but similar methods [11] also show at least some examples that are not human or only close to human shape.

Correctness: Yes.

Clarity: Yes.

Relation to Prior Work: Yes.

Reproducibility: Yes

Additional Feedback: - The authors should consider to upload their results in Table 2 of the appendix to the website of the MPI FAUST challenge at http://faust.is.tue.mpg.de/ (this can be done anonymously). I understand that the FAUST templates and scans are two separate datasets but judging from your results on the partial views, applying your network to the scans should be straightforward and it would be very instructive for the community. - Missing citation: The idea of modelling shape deformations in terms of locally rigid transformations is closely related to as-rigid-as-possible surface modelling [1]. In particular I believe that parts of the theory from Appendix D should be credited to [1]. - I would prefer to see the cumulative error curves for the correspondence accuracies instead of the average and recall values in Table 1 as they are a more expressive tool for comparing different results. - In your ablation study, I would like to see a comparison of the Laplacian embedding with the SHOT [2] descriptors which are the standard hand-crafted descriptor that is typically used in combination with deep architectures on 3D shapes. The HKS/WKS descriptors are purely intrinsic and not suitable for disambiguating symmetries etc. - I would suggest that the authors include qualitative examples that show a texture map between two full shapes. This is a standard tool to assess the quality of a matching between 3D shapes. [1] Sorkine, Olga, and Marc Alexa. "As-rigid-as-possible surface modeling." Symposium on Geometry processing. Vol. 4. 2007. [2] Salti, Samuele, Federico Tombari, and Luigi Di Stefano. "SHOT: Unique signatures of histograms for surface and texture description." Computer Vision and Image Understanding 125 (2014): 251-264.


Review 2

Summary and Contributions: This paper proposes a neural method to establish dense correspondences between between incomplete human scans and a template human model. The key idea is to obtain correspondences by finding local rigid transformations at each point in the input scan. The method has two components - an initialization step followed by a regularization step. The initialization step involves computing dense correspondences to the template using laplacian feature descriptors and finding local transformation based on these coarse correspondences. The regularization steps involves another neural network module which refines the initial coarse rigid transformations. The network design for the second module is inspired by a classical variational approach which involves the use of iteratively reweighted least squares

Strengths: The paper introduces three ideas: 1) It formulates the task of dense correspondence computation in terms of finding rigid transformations from the scan to template 2) It includes a robust theoretical analysis of the a classical transformation synchronization approach 3) It uses insights from 2) to formulate transformation synchronization in a neural module. This enables the entire pipeline of correspondence estimation to be trained in an end-to-end manner. While the idea of deforming shapes using local rigid transformations is not new (first introduced in Sumner et al. SIGGRAPH 04), to my knowledge, this is the first time where it is used for correspondence prediction. The authors show state-of-the-art or comparable results on three publicly available datasets - FAUST, SURREAL, SHREC19. The authors also perform ablation studies to show that the regularization module improves upon the initial correspondences. They also compare the neural network based regularization module with the classical regularization strategy and show the neural module performs better.

Weaknesses: The authors have only validated their method on human scans. It would have been more thorough to evaluate on meshes with different topologies. For example, they could have created a synthetic dataset of hands using the MANO model and evaluated their method on this dataset. Like [11] they could have trained their method on meshes generated using SMAL and used TOSCA animals for evaluation. I am not sure how practical the work is. People are usually scanned in clothing. The method is only evaluated on scans of people in minimal clothing. It would have been interesting to see whether, in addition to transformations due to pose and shape, the proposed method can also capture transformations due to clothing. The pipeline seems very complicated. I wonder how difficult it was to find all the hyperparameters to make the networks converge?

Correctness: The authors have used average correspondence error, 5cm, 10cm recall as metrics of choice. These are the standard metrics used for correspondence estimation.

Clarity: The paper is very well written and clearly structured. There are a few typos : eq 12) The variable \Phi is introduced in the minimization but is never mentioned before. eq 11) the targets are set to v_{c_i} but according to line 159 v \in R^12 and p is a point in R^3. It is not clear how the weights in eq 3) are initialized.

Relation to Prior Work: The related work is excellent. All major correspondence techniques are mentioned. The main prior works - SMPL registration, DHBC, 3D-CODED, DeepGeoFunc are all compared to, even though DeepGeoFunc was published at CVPR 2020

Reproducibility: Yes

Additional Feedback: The authors claim that the method is end-to-end trainable and yet in line 196 they mention that initial transformations are required for the regularization module. This could also be interpreted to mean that the initialization network and regularization network are trained separately. This ambiguity should be cleared. Figure 1 could be made cleaner by explicitly showing the two up-sampling and two down-sampling layers. The authors use the SURREAL dataset for training the method. AMASS - Mahmood et al. ICCV 2019- is a much larger dataset which should improve the proposed method. ----After rebuttal ----- I have read the rebuttal and the other reviews. The rebuttal addresses most of the points I had raised. The paper proposes a method with enough novelty to warrant an acceptance. I keep my score unchanged.


Review 3

Summary and Contributions: This paper studies the problem of establishing dense correspondences between partial scans of human models and a complete template model. The paper proposes a network-based framework to first initialize the correspondences, and then synchronize the transformations of different points to enforce smoothness and recover deformations. The network design is motivated by a principled optimization method.

Strengths: - The problem is well motivated. - The solution applies not only to 3D human body matching. It is a general method for 3D partial matching, although the network needs to be re-learned. - The experimental results are promising, and the ablation study is thorough.

Weaknesses: - It seems that the features descriptor is very important for the method. L137-138: Is the feature descriptor a one-to-one mapping? Is it possible that very different inputs could collapse into a cluster? Do we have a way to measure the robustness of the descriptor in this sense, so we are more confident to initialize the transformation using this descriptor? - In the objective function of the synchronization process (Eq 3), why don't we optimize the point cloud distance? I can see that the first term is optimizing that implicitly, but why not explicitly? - L165: what is the motivation to adopt iteratively reweighted least squares for optimization?

Correctness: The method seems to be correct.

Clarity: Overall it's reasonably clear, but it could be further improved. Currently Sec 3.2 and 3.4 are about network modules, while Sec 3.3 is about the principles that motivate the network. It reads a little fragmented.

Relation to Prior Work: Yes, it is discussed in details in the related work section.

Reproducibility: No

Additional Feedback: Although there are missing details about the implementation, it is not a concern since code is included in the supplementary materials. ========== After rebuttal ========== I read the rebuttal and comments from other reviewers, and I am leaning towards acceptance.


Review 4

Summary and Contributions: This work proposes a method to learn dense human correspondence in the case of partial scan to full scan (full-to-full also doable). The pipeline has two major components: 1. Initialization: using learnt per-point descriptors to drive weighted local rigid fitting. 2. Regularization (refinement): convert each step of a traditional optimization process into a recurrent graph network, while the alternating updates (eq. 5 and 6) are done by network operations. The regularization intuitively behaves like "as rigid as possible" constraints. The proposed method shows its performance in correspondence, and advantage over compared to sota methods.

Strengths: The design of the refinement (“regularization” as mentioned by the authors) contains some novelty. It also has connection to the traditional optimization steps and can be regarded as a neural network version of the optimization updates. From quantitative results, it shows the advantages over the previous methods.

Weaknesses: I would suggest the color mapping visualization for matching error (e.g. in Fig. 2 and 3) to be adjusted. The matching error is a non-negative scalar, therefore there is no need to use a diverging colormap that is intended for signed distance. It doesn’t visualize the error in a “perceptually linear” way. For example, the fig.2-right, it’s very hard to see the difference unless with extreme zoom-in. Maybe a “sequential” colormap such as “jet” or “hot”. See https://matplotlib.org/3.1.0/tutorials/colors/colormaps.html It seems like the “canonical” descriptor of the template mesh is provided by the laplacian embedding, and the learned descriptor ($f_{\Theta}(S)$) is “regularized” by the canonical descriptor. What happens when you learn the canonical descriptor freely? Maybe some ablation studies would be good to show the effect of this design. The representation for per-point transformation “v_i” is “scaled” rotation matrix elements and translation vectors. This appears a bit odd to me. First, if we operate directly on the elements, then the resulting matrix might not be a valid rotation. In this case, did you project it back to SO(3), or just allow it to be flexible so that the resulting transformation is outside of SE(3)? The recurrent network serves as a neural network version of iterative update as in the traditional optimization. Is the proposed 4x3=12 steps (4 graph layer, each with 3 alternations after) sufficient? It would be great to show results as each step is applied (to see their individual contribution). What if the graph network doesn’t do down-sampling and upsampling? Would it be possible to add discussion and comparison to recent point-cloud correspondence analysis, such as Fully Convolutional Geometric Features, Choy et al, ICCV 19? The Perfect Match: 3D Point Cloud Matching with Smoothed Densities, Gojcic et al., CVPR 19

Correctness: Mostly, except for there are some parts which are still not clear (see weakness session)

Clarity: Mostly yes, except for there are some parts which are still not clear (see weakness session)

Relation to Prior Work: I would add discussion on difference to recently point-cloud correspondence analysis papers such as: Fully Convolutional Geometric Features, Choy et al, ICCV 19 The Perfect Match: 3D Point Cloud Matching with Smoothed Densities, Gojcic et al., CVPR 19 These methods are tested on static scenes, but can be potentially used in dynamic scenarios such as human body.

Reproducibility: Yes

Additional Feedback: After rebuttal: After reading the rebuttal and comments from other reviewers, I think most of my concerns are addressed. Please address the issues and comments raised by all reviewers. I'm willing to increase my rating to 6: marginally accept.

[Author Response · NeurIPS 2020]

We are grateful for all comments and suggestions. Below we address questions raised by individual reviewers.

**Suggestions on adding or revising qualitative results and figures. (R1, R2, R4)**   We are thankful for these sugges-
tions and will add qualitative results on non-human shapes in the next version. This includes adding qualitative results
for non-human shapes and cumulative error curves for Table 1 and adjusting color mapping in Figures 2 and 3.

**Comparison to LES [9], DeepGeoFunc [10], 3DCODED [11] on Partial-to-full task. (R1)**   DeepGeoFunc requires
a triangular mesh as input, while 3DCODED and LES require a point cloud sampled from a full triangular mesh. In
other words, none of these methods ([9], [10], [11]) applies to partial depth scan input. Therefore, we only compared
our approach with these methods for the full-to-full task in the appendix (Table 2).

**Comparison to the SHOT descriptor. (R1)**   We used the code from the original SHOT paper to generate SHOT
descriptors for the template mesh vertices and used them to train our descriptor module for three days. The resulting
average correspondence errors on SURREAL/FAUST/SHREC19 are 5.97cm/6.66cm/11.04cm, respectively. Our results
(1.71cm/1.90cm/4.81cm) are significantly better.

**Investigating unsupervised setting. (R1)**   The major contribution of our paper is the transformation propagation
module, which learns from the error distribution of point-wise transformations to detect and rectify incorrect correspon-
dences that violate local rigidity. This strategy differentiates our approach from other methods based on hand-crafted
heuristics. Similar to other approaches, we can use the distance-preserving loss to train the transformation predictions.

**MPI FAUST Results. (R1)**   Most results that report on FAUST leaderboard use additional training data that signifi-
cantly impacted the results. In contrast, we compared all the methods under the same setup. We also plan to release the
code and data so that other methods can be compared under the same training/testing setup.

**Generalization from minimal clothing to clothing. (R2)**   Our approach is motivated by the piece-wise rigidity
assumption. Enforcing such an assumption is learned, meaning it can be adaptive to rigidity at different levels. For
clothing, it means more rigidity at the coarse level and less rigidity at the fine level.

**End-to-end Training. (R2)**   The descriptor module is pre-trained in order to generate initial transformations. Then
the whole pipeline (including the descriptor module) is trained end-to-end.

**Robustness of feature descriptors. (R3)**   We will add a visualization of the distribution of feature descriptors in the
revision. Note that the 10-cm recall of the correspondences derived from feature descriptors are above 95% for all
datasets. They provide a good foundation for the transformation synchronization module to rectify the error.

**Motivation of iteratively reweighted least squares (IRLS). (R3)**   IRLS is a popular method for solving regression
problems that involve robust objective functions, where the final solution is insensitive to a fraction of noisy measure-
ments. Suppose the noise ratio is below some constant (described in the main theorem). In that case, IRLS can be
proven to suppress such errors and converge to the underlying ground-truth. The paper's contribution is a learning
approach that leverages side information to reduce the input noise ratio (e.g., by reweighting) so that such a condition
holds.

**Ablation study on number of transformation propagation layers. (R4)**   The average correspondence error before
transformation propagation and after the first/second/third layer is 2.19cm/1.80cm/1.73cm/1.71cm for SURREAL,
2.59cm/1.97cm/1.89cm/1.90cm for FAUST, and 5.84cm/5.02cm/4.86cm/4.81cm for SHREC19. Most of the improve-
ment comes from the first layer. After the third layer, the effect of transformation propagation becomes marginal.

**Learn canonical descriptors freely. (R4)**   In our approach, canonical descriptors only provide initialization. We
first train our descriptor module to match Laplacian embedding descriptors. Then in the next phase of Training, the
descriptor module is fine-tuned with the transformation propagation module. Laplacian embedding descriptors worked
well. Since they are used only for the initialization, there was not much need to learn alternatives.

**What if no down-sampling/up-sampling. (R4)**   Down-sampling and up-sampling layers enable efficient computa-
tion. Empirically, these layers can provide 10x to 20x speedup without noticeable loss of performance.

**Transformation becomes invalid after interpolation. (R4)**   We allow the transformations after propagation to be
outside SE(3). This approach adds flexibility to encode local deformation. The same strategy was used in Sumner et al.
07. We will clarify this in the revision.

**Questions about equations. (R2, R3)**   The point distance version of Eq. 3 is much more complicated compared
to merely enforcing neighboring transformations to be equal. The weights are generated by a network that takes
transformations of neighboring points as input. In Eq. 11, $v_{c_i}$ should refer to a point on the template mesh. In Eq. 12, $\Phi$
should be removed.

[Meta-Review · NeurIPS 2020]

This submission proposes an approach to establishing dense correspondences between full and partial human body scans. It initially received four reviews with three positive scores (6,8,6,5). The reviewers appreciated strong performance, efficiency, novelty in the context of the given application and solid theoretical and empirical analysis. At the same time, reviewers would like to see the proposed method applied to other object categories with different topologies. The rebuttal addressed some of the remaining concerns, which resulted in an increase in scores to (6,8,6,6). For these reasons, the AC's recommendation is to accept this submission for presentation as a poster.